# Carbon-Supported High-Loading Sub-4 nm PtCo Alloy Electrocatalysts for Superior Oxygen Reduction Reaction

**DOI:** 10.3390/nano13162367

**Published:** 2023-08-18

**Authors:** Linlin Xiang, Yunqin Hu, Yanyan Zhao, Sufeng Cao, Long Kuai

**Affiliations:** 1School of Chemical and Environmental Engineering, Anhui Laboratory of Clean Catalytic Engineering, Key Laboratory of Production and Conversion of Green Hydrogen, Anhui Polytechnic University, Beijing Middle Road, Wuhu 241000, China; 17855321299@163.com (L.X.); 2221011456@ahnu.edu.cn (Y.H.); 2Institute of Energy, Hefei Comprehensive National Science Center, Hefei 230031, China; 3The Rowland Institute at Harvard, 100 Edwin H Land Blvd, Cambridge, MA 02142, USA; yzhao@rowland.harvard.edu; 4Aramco Boston Downstream Center, 400 Technology Square, Cambridge, MA 02139, USA; sufeng.cao@aramcoamericas.com

**Keywords:** carbon support, PtCo/C, fuel cells, electrocatalysis, high loading

## Abstract

Increasing the loading density of nanoparticles on carbon support is essential for making Pt-alloy/C catalysts practical in H_2_-air fuel cells. The challenge lies in increasing the loading while suppressing the sintering of Pt-alloy nanoparticles. This work presents a 40% Pt-weighted sub-4 nm PtCo/C alloy catalyst via a simple incipient wetness impregnation method. By carefully optimizing the synthetic conditions such as Pt/Co ratios, calcination temperature, and time, the size of supported PtCo alloy nanoparticles is successfully controlled below 4 nm, and a high electrochemical surface area of 93.8 m^2^/g is achieved, which is 3.4 times that of commercial PtCo/C-TKK catalysts. Demonstrated by electrochemical oxygen reduction reactions, PtCo/C alloy catalysts present an enhanced mass activity of 0.465 A/mg at 0.9 V vs. RHE, which is 2.0 times that of the PtCo/C-TKK catalyst. Therefore, the developed PtCo/C alloy catalyst has the potential to be a highly practical catalyst for H_2_–air fuel cells.

## 1. Introduction

Renewable energy technologies (such as solar energy, wind energy, hydropower, etc.) have developed rapidly in recent years [1,2,3]. However, due to their inherent intermittency, volatility, and randomness, they face the challenge of large-scale grid connection. Using hydrogen energy as a bridge to achieve the separation of renewable energy generation and terminal electricity consumption in terms of time is one of the best solutions for the large-scale application of renewable energy [4,5,6]. The primary approach is to use electricity generated from renewable energy to produce hydrogen and then continuously convert the chemical energy in hydrogen into electrical energy through proton exchange membrane fuel cells (PEMFCs) [7,8,9,10]. Unlike traditional thermal power generation, the hydrogen energy generation route using PEMFCs as carriers achieves zero emissions and has high energy conversion efficiency [11].

For electrochemical reactions in the cathode and anode of PEMFCs, the most efficient catalysts are precious metal platinum (Pt)-based catalysts [12,13,14,15]. However, due to the more complex cathodic oxygen reduction reaction (ORR) process and higher reaction energy barrier, it is necessary to use a large amount of Pt catalysts to achieve considerable performance. Therefore, developing high-performance Pt-based catalysts has always been a problem that continues to be solved to promote the large-scale commercialization of PEMFCs. Compared with Pt/C catalysts, using cheap transition metals (M) and Pt to construct alloy nanoparticles is currently widely considered the most promising method to solve the development problem of high-performance Pt-based catalysts [16,17,18,19,20]. A series of studies show that introducing other transition metals into the Pt lattice can regulate Pt’s electronic structure and improve Pt atoms’ utilization through ligand and geometric effects respectively, thus effectively improving Pt’s mass activity [21]. For example, Stamenkovic et al. first revealed that the specific activity of PtNi_3_ (111) single crystal catalysts is 90 times that of commercial Pt/C [22]. Shao-Horn’s research group developed PtNi alloy nanoparticles with a mass activity of ~0.6 A/mg_Pt_ at 0.9 V vs. RHE [23].

At present, many Pt-alloy/C catalysts reported in the literature have achieved high activity. However, the loading density of Pt-alloy nanoparticles is much lower than that of Pt/C catalysts [24,25,26]. This results in an increase in the thickness of the membrane electrode catalytic layer during practical applications and a proportional increase in the mass transfer impedance of O_2_, which limits the catalytic activity of the catalyst, especially when working at high current densities [27]. Generally, commercial fuel cell vehicle cathode catalysts require Pt loading greater than 50% (mass fraction) [27]. In addition, Pt-alloy formation requires high temperatures of 500–700 °C [28,29]. It can be imagined that high-density Pt-alloy particles are extremely prone to sintering at such high temperatures. Therefore, it is a great challenge to prepare small-sized Pt-alloy/C catalysts with a high loading of Pt as a practical catalyst for PEMFCs. Pt-alloy/C catalysts prepared in such conditions generally present a large size and low electrochemical surface area (ECSA). Herein, we present a 40% Pt-weighted sub-4 nm PtCo/C alloy catalyst via a simple large-scale incipient wetness impregnation method. By carefully optimizing the synthetic conditions such as Pt/Co ratios, calcination temperature, and time, the size of the supported PtCo nanoparticles is successfully controlled below 4 nm, and a high electrochemical surface area (ECSA) of 93.8 m^2^/g is achieved, which is 3.4 times that of commercial PtCo/C-TKK catalysts. Demonstrated by electrochemical oxygen reduction reactions, PtCo/C alloy catalysts present an enhanced mass activity of 0.465 A/mg at 0.9 V vs. RHE, which is 2.0 times that of the state-of-the-art commercial PtCo/C-TKK catalyst.

## 2. Materials and Methods

### 2.1. Materials’ Preparation

Typically, for preparing 40% Pt_3_Co_2_/C-8h catalysts, 0.84 g of 18.8% weighted H_2_PtCl_6_ solution and 0.32 mL of 1.27 mol/L Co(NO_3_)_2_·6H_2_O solution were added to 0.200 g of carbon black (Ketjenblack EC600JD). The mixture was dried in air at 200 °C for 2 h with a heating rate of 2 °C/min. Subsequently, the dried mixture was reduced at 560 °C in a 10 vol% H_2_/Ar with a heating rate of 2 °C/min. When cooling to room temperature, the powder was carefully transferred from the tube furnace.

The final Pt_3_Co_2_/C-8h alloy catalyst was obtained by etching the partial Co in a 1.0 mol/L H_2_SO_4_ solution. After etching, the products were washed three times and finally dried in a vacuum oven at 60 °C.

The other materials were obtained through similar processes. The differences were the Pt/Co ratio, the reduction temperature/time, or the kinds of cheap metals.

### 2.2. Materials’ Characterizations

The morphologies of samples were characterized via TEM (Technai G20 S-TWIN, FEI, Hillsboro, OR, USA) with a 200 kV acceleration voltage. The element distribution was studied using HAADF-STEM-EDX mapping. The phase of catalysts was analyzed using XRD (D8, Brook, Nordrhein-Westfalen, Germany) with Cu Kα radiation (λ = 0.15418 nm). The chemical states of surface elements were characterized via X-ray photoelectron spectroscopy (Thermo Fisher Scientific K-Alpha, ESCALAB 250XI, Waltham, MA, USA). Thermogravimetric analysis (STA, 2500) for the carbon content was conducted from 30 °C to 800 °C under an O_2_ atmosphere. The content of each element in the sample was analyzed by inductively coupled plasma atomic emission spectrometry (ICP-AES, Agilent 5110, Santa Clara, CA, USA).

### 2.3. Electrochemical Measurements

Rotating disk electrode (RDE) measurements were performed in a three-electrode electrochemical cell. The cell temperature was controlled with cycling water (25 °C). The reference electrode potentials were calibrated to the reversible hydrogen electrode (RHE) potentials. The working electrode was catalyst-modified RDE (5.0 mm) with a Pt loading of 15 µg cm^−2^. The catalysts were electrochemically activated by CV scanning between 0.05 and 1.05 V at 250 mV s^−1^ in a N_2_-saturated 0.1 M HClO_4_ solution. Then, the LSV curves were obtained from 0.05 V to 1.05 V at 10 mV s^−1^ in O_2_-saturated 0.1 M HClO_4_ with the rotating speed at 1600 rpm.

The mass-related kinetic current density (J_k_) of the catalyst at 0.9 V vs. RHE was calculated according to J_k_ = J × J_L_/((J_L_ − J) × L_Pt_), where J, J_L_, and L_Pt_ represent the current density at 0.9 V vs. RHE, the diffusion-limited current density, and Pt loading, respectively. The ohm resistance (R) was measured using electrochemical impedance spectroscopy from 0.1 Hz to 10,000 Hz with an amplitude of 5 mV at the initial potential of 0.05 V.

## 3. Results

### 3.1. Synthesis and Characterizations

The preparation process for PtCo/C alloy catalysts is described in the experimental section. Generally, a mixture solution containing H_2_PtCl_6_/Co(NO_3_)_2_ with a designated Pt/Co ratio was added to carbon black (Ketjenblack EC600JD). The loading of Pt was controlled to 40%; manually stir in the pan until the water is completely evaporated and the carbon black powder is uniformly dispersed in powder form. The dried H_2_PtCl_6_/Co(NO_3_)_2_/C mixture was subsequently reduced at 560 °C in 10% H_2_/Ar with a 50 mL/min flow rate. Other Pt-alloy catalysts synthesized with Cu, Fe, and Ni metals use the same synthetic method but change cobalt nitrate into other metal nitrates and adjust the calcination temperature for synthesis. Figure 1 shows a typical Pt_3_Co_2_/C-8h alloy catalyst obtained in a Pt/Co ratio of 3/2 and reducing the time to 8 h in 10% H_2_/Ar. All the catalysts were treated in 0.5 M H_2_SO_4_ before characterization and electrochemical measurements. According to the lower magnification transmission electron microscope (TEM) image (Figure 1a), PtCo alloy nanoparticles are uniformly dispersed on the carbon supports. It presents a small proportion of nanoparticles exceeding 6 nm. The higher magnification TEM image (Figure 1b) further shows partial small nanoparticles with a size of less than 2 nm. Based on the statistics of more than 200 random nanoparticles, the particle size of Pt_3_Co_2_/C-8h catalysts is 3.5 ± 1.0 nm. Furthermore, X-ray energy dispersive spectroscopy (EDS) mapping (Figure 1c) equipped with high-angle annular dark field scanning transmission electron microscopy (HAADF-STEM) exhibits a good overlap of Pt and Co elements in individual nanoparticles, suggesting a high alloy degree of Pt/Co.

We further investigated the effect of the Pt/Co ratio on the size and phase of alloyed nanoparticles. As shown in the X-ray powder diffraction (XRD) patterns (Figure 2a), all the diffraction peaks shifted after doping with Co in comparison with pure Pt (111) at 39.8°, and the peaks’ positions are between pure Pt and pure Co (111) at 42.6°. The shift in the XRD peaks to higher positions is also ascribed to the structural strain, namely compression stress, due to the smaller size of Co as compared with Pt [30]. As the Co content increases, the degree of shift increases. The more shifted degree indicates a higher alloy degree of Pt/Co [31]. However, the peak becomes sharper with the increase in Co content, indicating that the size of the supported nanoparticles becomes larger, which is well in agreement with the TEM image and size distribution. Pt_3_Co_1_/C-8h (Figure 2b,c) presents a uniform size distribution of 3.1 ± 0.9 nm. However, many large nanoparticles exist in Pt_1_Co_1_/C-8h (Figure 2d,e), suggesting that it is more difficult to control the size of highly alloyed catalysts.

As displayed in Figure 3, the calcination time in 10% H_2_/Ar also affects the size and phase of alloyed nanoparticles. XRD patterns (Figure 3a) show that the ordered intermetallic compounds appear within 4 h. When the reduction time is prolonged to 12 h, there is a significant increase in size. Typically, Pt_3_Co_2_/C-4h (Figure 3b,c) and Pt_3_Co_2_/C-12h (Figure 3d,e) possess a size distribution of 2.8 ± 0.8 nm and 3.9 ± 1.5 nm, respectively. In addition, we found that the diffraction peak of Pt_3_Co_2_/C-4h is asymmetric, indicating that there are partial PtCo nanoparticles with a low degree of alloying. Thus, the size and degree of alloying are a pair of constraints.

We further explored the carbon-supported 40% Pt-alloy catalysts with the cheaper Cu, Fe, and Ni metals. According to the HAADF-STEM-EDS mappings, the signal of Pt is well overlapped with the corresponding Cu (Figure 4a), Fe (Figure 4b), and Ni (Figure 4c) elements, indicating the formation of highly alloyed nanoparticles [32]. However, there were significant differences in the size distribution. For Pt_3_Cu_2_/C, we can obtain a highly uniform size distribution of 3.1 ± 0.9 nm (Appendix A). However, the size control of PtFe/C (Appendix A) and PtNi/C (Appendix A) is still a big challenge.

### 3.2. Electrochemical Evaluation of the Oxygen Reduction Reaction

The electrochemical surface area (ECSA) and mass activity at 0.9 V (vs. RHE, and the same below) (MA_0.9_) of 40% PtCo/C alloy catalysts were measured on a rotating disc electrode (RDE). To obtain the ECSAs, the cyclic voltammetry (CV) curves (Figure 5a) were carried out in N_2_-saturated 0.1 M HClO_4_. The ECSAs were calculated according to the hydrogen adsorption/desorption current densities [33]. To obtain the MA_0.9_ values, the linear sweep voltammetry (LSV) curves (Figure 5b) were performed in O_2_-saturated 0.1 M HClO_4_ and corrected with an ohmic resistance of 23 Ω determined by electrochemical impedance testing. Figure 5c,d displays the Pt/Co ratio-dependent ECSA and MA_0.9_ values. The ECSAs of Pt_3_Co_1_/C-8h, Pt_3_Co_2_/C-8h, and Pt_1_Co_1_/C-8h are 93.9, 86.8, and 26.1 m^2^/g, respectively. The MA_0.9_ of Pt_3_Co_1_/C-8h, Pt_3_Co_2_/C-8h, and Pt_1_Co_1_/C-8h are 0.292, 0.354, and 0.273 A mg^−1^. It can be seen that the ECSA of Pt_3_Co_1_/C-8h is slightly higher than that of Pt_3_Co_2_/C-8h, but the MA_0.9_ is lower. High ECSA corresponds to more electrochemically active sites that have smaller sizes, while high MA_0.9_ not only requires smaller particle sizes but also a better electronic structure of nanoalloys. As shown in Figure 2a, the diffraction peak of Pt_3_Co_1_/C-8h is wider than that of Pt_3_Co_2_/C-8h. According to the Scherrer formula, the local size of Pt_3_Co_1_/C-8h is smaller, but the diffraction peak shift of Pt_3_Co_1_/C-8h is smaller than that of Pt_3_Co_2_/C-8h, indicating that the degree of the alloy significantly affects the activity. In addition, Pt_1_Co_1_/C-8h with the highest alloy degree exhibits lower ECSA and lower MA_0.9,_ which are caused by the larger particle sizes [32]. Thus, we must coordinate alloying degree and size to achieve optimal performance.

As shown in Figure 6a, the CV curve shows that the hydrogen adsorption/desorption current density of the prepared Pt_3_Co_2_/C-4h catalyst (black curve) is higher than that of both Pt_3_Co_2_/C-8h and Pt_3_Co_2_/C-12h catalysts, indicating that the ECSA is larger. In addition, according to Figure 6b, the LSV curves of both the Pt_3_Co_2_/C-4h and Pt_3_Co_2_/C-8h catalysts show a larger polarization current at 0.9 V than Pt_3_Co_2_/C-12h. As shown in Figure 6c, the ECSA values for Pt_3_Co_2_/C-4h, Pt_3_Co_2_/C-8h, and Pt_3_Co_2_/C-12h are 93.8, 62.7, and 60.4 m^2^/g, respectively. The MA_0.9_ values (Figure 6d) for Pt_3_Co_2_/C-4h, Pt_3_Co_2_/C-8h, and Pt_3_Co_2_/C-12h are 0.465, 0.402, and 0.341 A/mg, respectively. As shown in Figure 6e, Pt_3_Co_2_/C-4h has a lower Tafel slope of 59.3 mV dec-1, indicating that it has a faster kinetic process. In addition, we also calculated the specific activity (SA) of the catalyst in Figure 6f. Higher SA means a higher conversion frequency of a single active site, and rapid mass transfer will also accelerate the dissolution of Pt atoms while increasing the pressure of material transport [34]. Therefore, our work is mainly devoted to the construction of small-sized alloy nanocatalysts with higher ECSA and MA_0.9_. We further studied the stability of the Pt_3_Co_2_/C-4h catalyst as shown in Appendix A. Under the accelerated durability test (ADT) of 10,000 cycles, the ECSA loss rate was 14.8% and the MA_0.9_ loss rate was 32.2%. XRD patterns (Figure 3a) have shown that the Pt_3_Co_2_/C-4h catalyst possesses an optimized alloying degree and size distribution, so it presents the highest MA_0.9_ value.

The surface chemical composition and elemental valence of each element in the prepared samples were characterized using X-ray photoelectron spectroscopy (XPS). As shown in (Figure 6g,h), the signal of surface Co is very weak, and the XPS detection depth is several atomic layers on the surface, indicating that after acid etching, the surface of the alloy nanoparticles is dominated by Pt. In addition, Pt in the Pt 4f region and Co in the Co 2p region exist in the metal states of (Pt^0^) and (Co^0^). Compared with the standard Pt^0^ 4f peak (71.13 eV and 74.40 eV), Pt 4f_7/2_ and Pt 4f_5/2_ show lower binding energies (70.83 eV and 74.13 eV). Figure 6h shows that the binding energy of the Co^0^ 2p_3/2_ and Co^0^ 2p_1/2_ peaks (780.33 eV and 795.4 eV) has a positive shift in comparison with the standard Co^0^ 2p peaks (777.80 eV and 793.30 eV). These changes in binding energy indicate that there is an electronic interaction between Pt and Co.

We also carried out thermogravimetric analysis (TGA) for the carbon content in the sample of Pt_3_Co_2_/C-4h. TGA was conducted from 30 °C to 800 °C under an O_2_ atmosphere. As shown in Appendix A, about 5% of the mass loss before 100 °C is caused by the adsorbed water in the sample [35]. At first, the mass of the carbon matrix decreases slowly with the increase in temperature. After the temperature of about 344 °C, the sample curve depicts obvious weight loss, which can be attributed to the complete oxidation of the carbon matrix; the weight loss of Pt_3_Co_2_/C-4h is 52.5%. Therefore, the metal loading can be calculated to be about 44.5%. To illustrate the atomic ratio of Pt_3_Co_2_/C-4h metal, we carried out an ICP test. The ratio of Pt to Co after acid etching is 1.46:1, which is basically in line with our expectations.

As a meaningful addition, we also studied the electrochemical performance of the obtained catalysts, as shown in Figure 4. The CV curves (Figure 7a) show that the hydrogen adsorption/desorption current density of Pt_3_Cu_2_/C (black curve) is higher than that of Pt_3_Co_2_/C, Pt_3_Fe_2_/C, and Pt_3_Ni_2_/C catalysts. The LSV curve of the Pt_3_Co_2_/C catalyst shows the largest polarization current at 0.9 V. The ECSA values (Figure 7c) for Pt_3_Cu_2_/C, Pt_3_Co_2_/C, Pt_3_Ni_2_/C, and Pt_3_Fe_2_/C catalysts are 102, 93.8, 70.8, and 55.2 m^2^/g, respectively. The MA_0.9_ values (Figure 7d) for Pt_3_Cu_2_/C, Pt_3_Co_2_/C, Pt_3_Ni_2_/C, and Pt_3_Fe_2_/C are 0.386, 0.465, 0.407, and 0.257 A/mg, respectively. It can be found that the size distribution and activity can be further adjusted by changing the doping metals.

## 4. Discussion

Above all, we demonstrate a simple incipient wetness impregnation method to obtain a series of small-sized PtM (M = Fe, Co, Ni, and Cu) nanoalloy catalysts with high loadings. The whole process is carried out at room temperature and pressure to ensure that the active component is fixed to the carbon carrier to the maximum extent during the solvent evaporation process. The optimized 40% weighted PtCo/C electrocatalyst is Pt_3_Co_2_/C-4h with ECSA and MA_0.9_ values of 93.8 m^2^/g and 0.465 A/mg, which are 3.4 and 2.0 times that of commercial PtCo/C catalysts (TANAKA TKK, Appendix A). The Pt_3_Co_2_/C-4h catalyst shows superior application potential for PEMFCs. In addition, we also find that the Pt_3_Ni_2_/C catalyst is a highly active catalyst with an MA_0.9_ value of 0.407 A/mg. As a potential catalyst, its MA_0.9_ value would be further enhanced if the particle size could be well controlled. However, it is more challenging to synthesize high-loading PtNi/C catalysts with a size lower than 4 nm.

## 5. Conclusions

In summary, this work presents a 40% Pt-weighted sub-4 nm PtCo/C alloy catalyst via a simple incipient wetness impregnation method. The optimized PtCo/C alloy catalyst achieves a high electrochemical surface area of 93.8 m^2^/g. Demonstrated by electrochemical oxygen reduction reactions, PtCo/C alloy catalysts present a mass activity of 0.465 A/mg at 0.9 V vs. RHE, which is 2.0 times that of the PtCo/C-TKK catalyst. The developed PtCo/C alloy catalysts have the potential to be highly practical catalysts for H_2_-air fuel cells.

## Figures and Tables

**Figure 1 nanomaterials-13-02367-f001:**
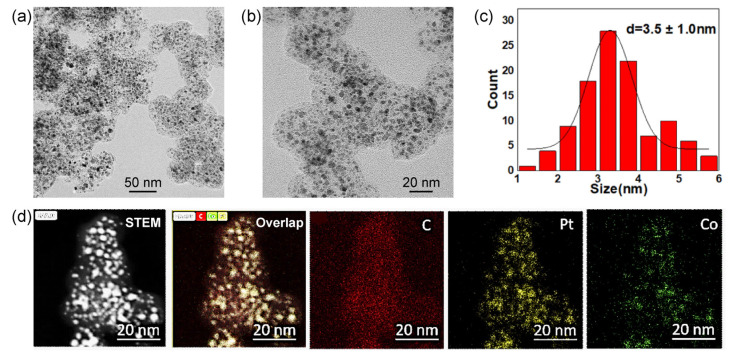
Lower (**a**) and higher (**b**) magnification TEM images, size distributions of supported nanoparticles (**c**), and HAADF-STEM-EDS-mappings (**d**) of Pt_3_Co_2_/C-8h catalysts with a Pt loading of 40%. The subfigures from left to right in (**d**) mean HAADF-STEM image, overlapped image C, Co and Pt images.

**Figure 2 nanomaterials-13-02367-f002:**
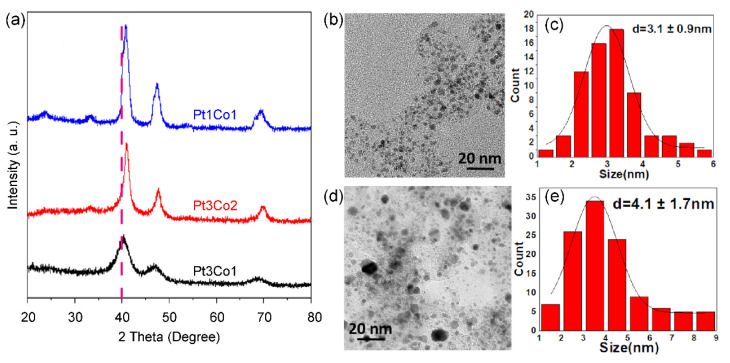
XRD patterns of (**a**) Pt_3_Co_1_/C-8h (black curve), Pt_3_Co_2_/C-8h (red curve), and Pt_1_Co_1_/C-8h (blue curve) catalysts. TEM images (**b**,**d**) and size distributions (**c**,**e**) of supported nanoparticles of Pt_3_Co_1_/C-8h (**b**,**c**) and Pt_1_Co_1_/C-8h (**d**,**e**) catalysts.

**Figure 3 nanomaterials-13-02367-f003:**
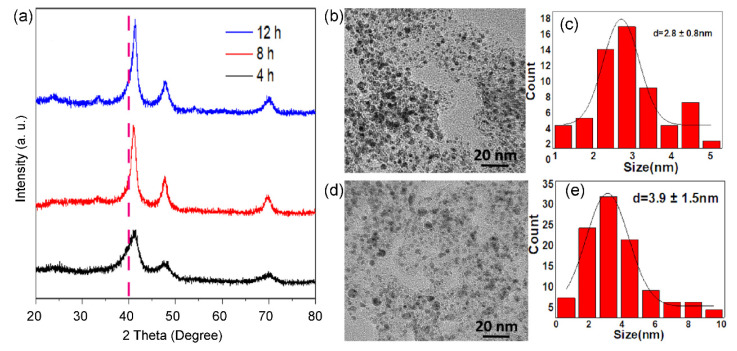
XRD patterns (**a**) of Pt_3_Co_2_/C-4h (black curve), Pt_3_Co_2_/C-8h (red curve), and Pt_3_Co_2_/C-12h (blue curve) catalysts. TEM images (**b**,**d**) and size distributions (**c**,**e**) of supported nanoparticles of Pt_3_Co_2_/C-4h (**b**,**c**) and Pt_3_Co_2_/C-12h (**d**,**e**) catalysts.

**Figure 4 nanomaterials-13-02367-f004:**
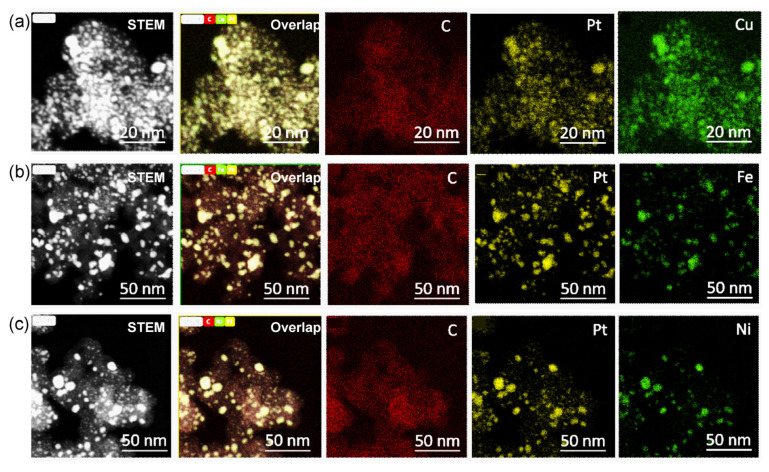
HAADF-STEM-EDS-mappings of Pt_3_Cu_2_/C (**a**), Pt_3_Fe_2_/C (**b**), and Pt_3_Ni_2_/C (**c**) catalysts with a Pt loading of 40%. The subfigures from left to right in (**a**) mean HAADF-STEM image, overlapped image C, Cu and Pt images, and it is similar for (**b**,**c**).

**Figure 5 nanomaterials-13-02367-f005:**
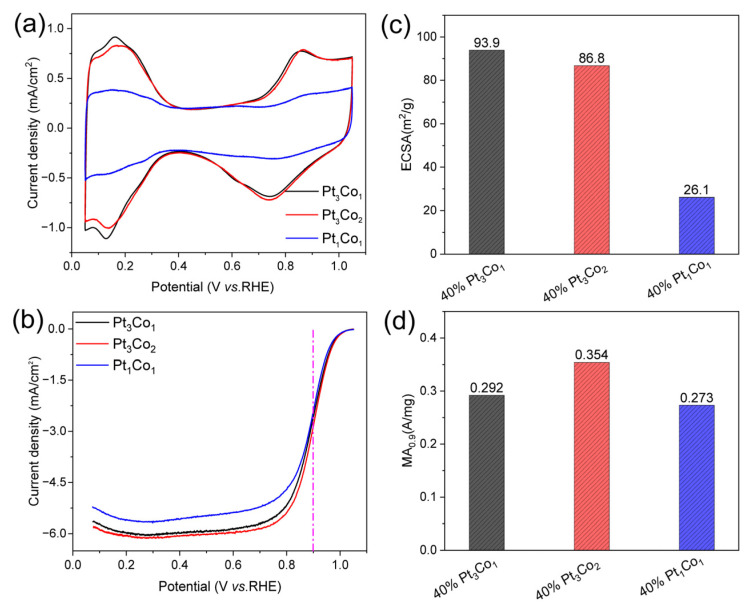
Electrocatalytic performance. (**a**) CV curves in N_2_-saturated 0.1 M HClO_4_, (**b**) LSV curves of the Pt_3_Co_1_/C-8h, Pt_3_Co_2_/C-8h, and Pt_1_Co_1_/C-8h catalysts (with internal resistance corrected) in O_2_-saturated 0.1 M HClO_4_. Comparison of (**c**) ECSAs and (**d**) MAs of the catalysts at 0.9 V (vs. RHE). The measurements were performed on an RDE with a rotating speed of 1600 rpm at 25 °C.

**Figure 6 nanomaterials-13-02367-f006:**
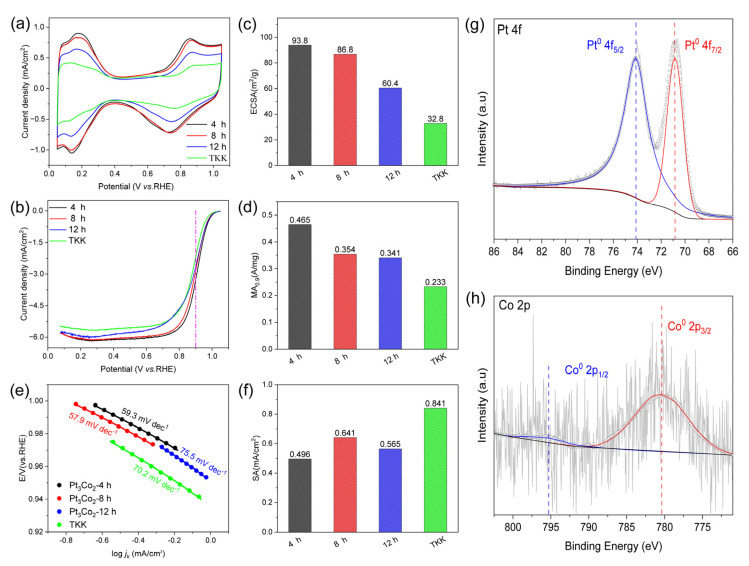
Electrocatalytic performance. (**a**) CV curves in N_2_-saturated 0.1 M HClO_4_, (**b**) LSV curves of the Pt_3_Co_2_/C-4h, Pt_3_Co_2_/C-8h, Pt_3_Co_2_/C-12h, and TKK catalysts (with internal resistance corrected) in O_2_-saturated 0.1 M HClO_4_. Comparison of (**c**) ECSAs and (**d**) MAs of the catalysts at 0.9 V (vs. RHE). (**e**) Tafel curves. (**f**) Specific activities (SAs). (**g**,**h**) High-resolution XPS results of Pt 4f and Co 2p for Pt_3_Co_2_/C-4h particles. The measurements were performed on an RDE with a rotating speed of 1600 rpm at 25 °C.

**Figure 7 nanomaterials-13-02367-f007:**
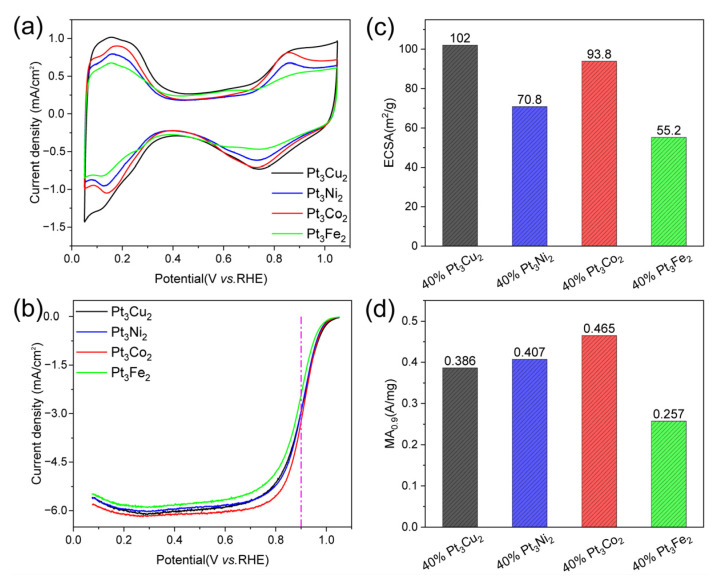
Electrocatalytic performance. (**a**) CV curves in N_2_-saturated 0.1 M HClO_4_, (**b**) LSV curves of the Pt_3_Cu_2_/C, Pt_3_Ni_2_/C, Pt_3_Co_2_/C, and Pt_3_Fe_2_/C catalysts (with internal resistance corrected) in O_2_-saturated 0.1 M HClO_4_. Comparison of (**c**) ECSAs and (**d**) MAs of the catalysts at 0.9 V (vs. RHE). The measurements were performed on an RDE with a rotating speed of 1600 rpm at 25 °C.

## Data Availability

All data presented in the article are available upon request.

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
