# Peer review of "Carbon-Supported High-Loading Sub-4 nm PtCo Alloy Electrocatalysts for Superior Oxygen Reduction Reaction"

_nanomaterials, 2023, doi:10.3390/nano13162367_

Round 1

Reviewer 1 Report

This paper focused on improving the ORR activity of PtCo/C alloy catalysts with high loadings though optimizing the synthetic conditions such as Pt/Co ratios, calcination temperature and time, the size of supported PtCo alloy nanoparticles. However, some descriptions are unclear, and more evidence is needed to support the statements in this paper. I would recommend below comments to be addressed properly before acceptance of publication on Nanomaterials.

1.Why is a scanning rate of 250 mV s-1 used for CV measurement? Is this a good scanning rate for capturing all the red-ox feature of the catalysts and for accurate ESCA calculation?

2. Does the 40% Pt-weighted loading come from the precursors or from the final product? What’s the Pt: Co ratio from STEM-EDS results for the catalysts measured in this paper? Are they as expected?

3. How does calcination time affect Pt:Co ratio? If yes, what adjustments are needed?

4. In line 174, the authors stated that ‘the degree of alloy significantly affects the activity’. Please add more explanation, discussion and supporting refs.

5. Some typos need to be corrected:

a. Line 172-173, ‘The MA0.9 of Pt3Co1/C-8h, Pt3Co1/C-8h and Pt1Co1/C-8h are 0.0.284, 0.393 and 0.275 A mg-1.

b. Line 213, ‘Pt3Co2/C-4h catalyst shows superior application potential for DMFCs.’

c. Line 218, ‘In summary, this work presents a 40% Pt-weighted sub-4 nm PtCo/C alloy catalysts’

5. CV curve for Pt3Co2 seems to be different in Figure 5a and 7a. Please double-confirm and explain the difference.

6. It’s hard to conclude that the PtCo/C-based catalysts as described and studied in this paper is a good candidate for H2-Air fuel cell. Please refine the intro and conclusion parts.

Some typos exist. Some sentences can be better written for a clearer statement.

Author Response

Great thanks for your careful review, positive appraisal and valuable comments on our work. By addressing all your issues, we have carefully revised our manuscript for your consideration.

Please see the attachment for the details of response to Reviewer 1.

Reviewer 2 Report

The manuscript describes the synthetic conditions for obtaining of the  PtCo alloy electrocatalysts demonstrating high activity in the oxygen  reduction reaction. The observed activity exceed that for commercial catalyst of the same type; this is very good result. The effect of Pt/Co ratio on the size and phase of alloyed nanoparticles was investigated.  Optimization and characterization of new catalysts were perfectly done; the results obtained seem to be quite reliable.

However, there are some points that should be addressed prior to publication of the paper.

1.      Fig.6c: the non-monotonous changing in the ECSA values for Pt3Co2/C-4h, Pt3Co2/C-8h and Pt3Co2/C-12h catalysts seems strange.  Additional comments clarifying the reason would be useful.

2.      Linear sweep voltammetry (LSV) rather than linear cyclic voltammetry (p.5)

3.      The LSV curve for the commercial  PtCo/C catalysts (TANAKA TKK) in O2-saturated solution should be included in Fig.6a for comparison

4.      The Discussion section is too short. Two sentences about Co-containing catalysts are given and two sentences about the Ni counterpart. It is not enough. More information about structure-property relationships should be provided; especially, about Ni which is poorly discussed in the text.

Minor editing of English language is requied

Author Response

Thanks very much for taking your time to review this manuscript. I really appreciate all your comments and suggestions! Please find my itemized responses in below and my revisions in the re-submitted files.

Reviewer 3 Report

What is the novelty of the work? A large number of authors use a similar synthesis method to obtain bimetallic catalysts. It is necessary to indicate works with a similar synthesis method in the literature review and emphasize the features of this work.

The description of the synthesis requires clarification: for example, how long was the mixture of salts and carbon support stirred before heating, was ultrasonic homogenization used, how the material was cooled after heat treatment, which salts were used in the case of the Pt-alloy catalysts synthesis with the Cu, Fe, and Ni metal.

It is necessary to determine the materials composition before and after etching in acid.

It is necessary to confirm the metals mass fraction in catalysts by TGA/DSC.

The XPS method must be used to determine the surface composition.

page 3, line 127 "pure Pt (36.80)" - position of platinum reflection is incorrect.

the authors mention in the article (line 129 and 194) the intermetallic phase, while X-ray diffraction patterns do not show superstructural reflections - therefore, it is incorrect to speak of the Pt-Co intermetallic compounds presence, apparently an alloy is obtained.

It is necessary to estimate the crystallites size according to the Scherrer equation according to XRD data and compare these data with the NPs size according to TEM data.

It is necessary to calculate the solid solution composition according to Vegard's law, compare it with the composition of the obtained catalysts, and determine the degree of metals alloying in nanoparticles according to XRD data.

The authors should compare the activity of the catalysts in Tafel coordinates and compare with the commercial one, determine the Tafel slopes for the obtained catalysts.

The authors use mass activity (on the mass of platinum), but it is necessary to additionally use and analyze the specific activity (on the ESA value).

in Figure 7a, the curve for Pt3Co2 is significantly different, the reason must be specified.

line 172 - typo "0.0.284"

line 213 - typo "DMFCs" - most likely meant PEMFCs

Author Response

Thank you for your comments. The introduction part of our original paper does not write the new idea and importance clearly; in view of this, we have strengthened the introduction to emphasize innovation.

Please see the attachment for the details of response to Reviewer 1.

Reviewer 4 Report

In this work, the authors used a simple incipient wetness impregnation method to prepare high-loading sub-4 nm PtCo alloy electrocatalysts for superior ORR. Some suggestions are helpful to improve the quality of this manuscript. If the authors well solve these comments, the manuscript can be accepted.

1.      In figure 2a, the shift of XRD peaks to higher positions is also ascribed to the structural strain, namely compression stress, duo to the smaller size of Co as compared with Pt. The author should well mention this and refer to this work (10.1063/5.0083059).

2.      The related descriptions below and about Figures 5a-d and 6a-d are chaotic. Please well check and revise these parts.

3.      Necessary ORR stability should be measured, for example chronoamperometric curves. Please refer to this work with DOI of 10.1002/chem.201705675.

4.      Necessary performance comparisons should be given, such as 10.1016/j.compositesb.2021.109082.

Minor editing of English language required

Author Response

Great thanks for your interest and positive evaluation on our work! We are very grateful and willing to accept all your valuable comments and suggestions to further improve this work. By addressing all your suggestion, we have carefully revised our manuscript for your consideration.

Please see the attachment for the details of response to Reviewer 4.

Round 2

Reviewer 1 Report

After revision, the results and discussion were presented in a clearer and more reasonable way. Many mistakes and typos were corrected. This paper can be accepted for publication on Nanomaterials.

Reviewer 3 Report

The authors responded to the comments and made the necessary corrections.